# CRISPR/Cas9-Mediated Deletion of Large Genomic Fragments in Soybean

**DOI:** 10.3390/ijms19123835

**Published:** 2018-12-01

**Authors:** Yupeng Cai, Li Chen, Shi Sun, Cunxiang Wu, Weiwei Yao, Bingjun Jiang, Tianfu Han, Wensheng Hou

**Affiliations:** 1National Center for Transgenic Research in Plants, Institute of Crop Sciences, Chinese Academy of Agricultural Sciences, Beijing 100081, China; caiyupeng2015@126.com (Y.C.); chenli01@caas.cn (L.C.); yaoweiwei_nky@126.com (W.Y.); 2Ministry of Agriculture Key Laboratory of Soybean Biology (Beijing), Institute of Crop Sciences, Chinese Academy of Agricultural Sciences, Beijing 100081, China; sunshi@caas.cn (S.S.); wucunxiang@caas.cn (C.W.); jiangbingjun@caas.cn (B.J.); hantianfu@caas.cn (T.H.)

**Keywords:** *Agrobacterium tumefaciens*-mediated transformation, CRISPR/Cas9, dual-sgRNA/Cas9 design, large fragment deletion, soybean

## Abstract

At present, the application of CRISPR/Cas9 in soybean (*Glycine max* (L.) Merr.) has been mainly focused on knocking out target genes, and most site-directed mutagenesis has occurred at single cleavage sites and resulted in short deletions and/or insertions. However, the use of multiple guide RNAs for complex genome editing, especially the deletion of large DNA fragments in soybean, has not been systematically explored. In this study, we employed CRISPR/Cas9 technology to specifically induce targeted deletions of DNA fragments in *GmFT2a* (Glyma16g26660) and *GmFT5a* (Glyma16g04830) in soybean using a dual-sgRNA/Cas9 design. We achieved a deletion frequency of 15.6% for target fragments ranging from 599 to 1618 bp in *GmFT2a*. We also achieved deletion frequencies of 12.1% for target fragments exceeding 4.5 kb in *GmFT2a* and 15.8% for target fragments ranging from 1069 to 1161 bp in *GmFT5a*. In addition, we demonstrated that these CRISPR/Cas9-induced large fragment deletions can be inherited. The T2 ‘transgene-free’ homozygous *ft2a* mutants with a 1618 bp deletion exhibited the late-flowering phenotype. In this study, we developed an efficient system for deleting large fragments in soybean using CRISPR/Cas9; this system could benefit future research on gene function and improve agriculture via chromosome engineering or customized genetic breeding in soybean.

## 1. Introduction

In recent years, the CRISPR/Cas9 (clustered regularly interspaced short palindromic repeat / CRISPR-associated protein 9) system has been rapidly adopted and widely used for genome editing in various organisms owing to its advantages of simplicity, efficiency, cost-effectiveness, and versatility and the ability to simultaneously target multiple genes within a cell [1]. The CRISPR/Cas9 system requires three key components: Cas9 endonuclease (it comprises two nuclease domains: the RuvC-like domain and HNH domain), trans-activating crRNA (tracrRNA), and a precursor crRNA (pre-crRNA) array containing nuclease guide sequences [2]. The Cas9 endonuclease can be guided by a synthetic single-guide RNA (sgRNA) to recognize target sequences [3] and create double-strand breaks (DSBs) at specific genomic loci, which are often repaired subsequently by DNA self-repair mechanisms in the cell through non-homologous end joining (NHEJ) and/or homology-directed repair (HDR) pathways [4]. The NHEJ pathway is error-prone and typically generates some base insertions/deletions (indels) within the target sequence. These indels introduce frameshift mutations in the protein-coding regions of genes of interest or disrupt important functional domains, often resulting in gene inactivation [5,6]. In addition, multiple DSBs can be utilized to mediate larger fragment deletions within a single genome [2,4]. This purpose can be achieved using dual-sgRNA targeting two loci on the same chromosome [7]. Large fragment deletions between two nuclease-targeted loci have been reported in various non-plant species, such as human [2,7,8,9], zebrafish [10], *Saccharomyces cerevisiae* [11], and mouse [12,13,14], and this method has great importance for illustrating the function and regulatory elements of genes [15]. Some studies have demonstrated that the CRISPR/Cas9 system can effectively edit DNA fragments ranging from dozens of bases to greater than 1 Mb, which suggest that this method can be utilized to operate arbitrary lengths of DNA fragments at many optional positions in the genome [12,16,17]. The CRISPR/Cas9 system has also been applied to produce large fragment deletions in plants [18], although this application has been relatively limited compared with that in non-plant species. For instance, large chromosomal deletions (115–245 kb) containing three different clusters of genes in rice protoplasts and two clusters in regenerated T0 generation plants were successfully produced [19]. Another study achieved frequencies of up to 21% for a 430 bp fragment deletion and 9% for a 10 kb fragment deletion in *DEP1* in *Indica* rice using the CRISPR/Cas9 editing system [20]. A combinatory dual-sgRNA/Cas9 vector was successfully utilized to delete miRNA gene regions (*MIR169a* and *MIR827a*) in *Arabidopsis thaliana* [21]. A newly developed CRISPR/Cas9-based toolkit was exploited for the induction of inheritable genomic fragment deletions, and its multiplexing capacity was demonstrated at six different loci in *Nicotiana benthamiana* and *Arabidopsis* [22]. Multiple site targeting was achieved in cotton using several sgRNAs simultaneously, generating targeted gene fragment deletions in the *GhPDS* locus [23]. Heritable genomic fragment deletions were produced in the putative barley *ENGase* gene [24]. These aforementioned cases demonstrate the power of CRISPR/Cas9 for genomic fragment deletions in plants and its potential applications in further studies.

The earliest reports of genome editing in soybean (*Glycine max* (L.) Merr.) using CRISPR/Cas9 date from 2015. The abilities of this new genome editing system to generate and estimate targeted mutagenesis in both endogenous and exogenous genes have been demonstrated in soybean hairy roots [25,26,27,28,29,30] and whole plants from embryonic calluses transformed by particle bombardment [31]. Targeted gene integrations induced by HDR were also detected using border-specific PCR analysis at the callus stage, which revealed that one HDR event was transmitted to the T1 generation [31]. *Rj4*, not the gene previously reported, was finely mapped, and, through the use of both complementation tests and CRISPR/Cas9-based gene knockout experiments, it was demonstrated that this gene has evolved the ability to control nodulation specificity in soybean [30]. In a previous study, we employed the CRISPR/Cas9 system to specifically induce site-directed mutations of the *GmFT2a* gene, a key flowering integrator in soybean, and found that the homozygous *ft2a* mutants exhibited late flowering under both long-day and short-day conditions. We also obtained some homozygous ‘transgene-free’ *ft2a* mutants for more in-depth research [32]. To summarize, these aforementioned cases have shown that CRISPR/Cas9 is a simple, efficient, and robust technology for genome editing in soybean. At present, this genome-editing tool has been mainly focused on the knock-out of targeted genes, and most site-directed mutagenesis mediated by CRISPR/Cas9 has occurred at single cleavage sites with short deletions and/or insertions owing to the use of the single-sgRNA/Cas9 system in soybean. However, the use of multiple guide RNAs for complex genome editing in soybean, especially large DNA fragment deletion, has not been systematically explored. 

In this study, we employed CRISPR/Cas9 technology to specifically induce targeted deletions of DNA fragments in two *FLOWERING LOCUS T* (*FT*) homologs, *GmFT2a* (Glyma16g26660) and *GmFT5a* (Glyma16g04830), in soybean using a dual-sgRNA/Cas9 design. We constructed five vectors targeting single sites (three for *GmFT2a*, two for *GmFT5a*) and three combinatory dual-sgRNA/Cas9 vectors carrying the corresponding pairs of sgRNAs. We transformed all eight constructs into the soybean cultivar Jack by *Agrobacterium tumefaciens*-mediated transformation and tested the efficiencies of mutations at single target sites and simultaneous mutations at both target sites among the T0 plants. We achieved large fragment deletions between pairs of target sites successfully and confirmed the heritability of these mutations. The T2 ‘transgene-free’ homozygous *ft2a* mutants we obtained exhibited the late-flowering phenotype. Our results indicate that the CRISPR/Cas9 system is a powerful tool for the robust and dependable generation of genomic fragment deletions in soybean.

## 2. Results

### 2.1. Experimental Design of Targeted Fragment Deletions of GmFT2a and GmFT5a

In the present study, we selected three sgRNAs targeting different sites in *GmFT2a*: *GmFT2a*-D1, *GmFT2a*-D2, and *GmFT2a*-D3. These three target sites are located in the first, third, and fourth (last) exons of *GmFT2a* (Figure 1A,B). The fragment between the cleavage sites in *GmFT2a*-D1 and *GmFT2a*-D2 is 1173 bp, whereas that between the cleavage sites in *GmFT2a*-D1 and *GmFT2a*-D3 is 4556 bp. We also designed two sgRNAs targeting different sites in *GmFT5a*; these sites are named *GmFT5a*-D1 and *GmFT5a*-D2 and are located in the first and fourth (last) exons of *GmFT5a* (Figure 1C). The fragment between the cleavage sites in *GmFT5a*-D1 and *GmFT5a*-D2 is 1070 bp. We constructed five vectors targeting the aforementioned single sites to induce site-specific mutations. To obtain targeted fragment deletions in *GmFT2a* and *GmFT5a*, three combinatory dual-sgRNA/Cas9 vectors—*GmFT2a*-D1/D2, *GmFT2a*-D1/D3, and *GmFT5a*-D1/D2—were used to delete the DNA fragments between each pair of sgRNAs (Figure 1D). All eight constructs were transformed into the soybean cultivar Jack using *Agrobacterium tumefaciens*.

### 2.2. Detection of Site-Specific Mutations and Large Fragment Deletions

Generally, a new technology with a wide range of applications has higher requirements for efficiency, so we examined the frequencies of site-specific mutations at single target sites among the T-DNA-positive T0 lines. Genomic DNA was extracted from the leaves, and PCR was performed with specific primers to amplify the genomic regions containing the target sites (Figure 1). Sequencing analysis was performed to detect mutations. The frequencies of site-specific mutations were calculated by dividing the number of T0 plants with any type of mutation at a single target site by the total number of T0 events harboring the corresponding T-DNA of the sgRNA/Cas9 vectors. As shown in Table 1, among the T0 lines transformed by constructs targeting single sites, 48.0% (12 of 25), 52.4% (11 of 21), and 72.2% (13 of 18) of the T0 lines had heterozygous targeted mutations at the *GmFT2a*-D1, *GmFT2a*-D2, and *GmFT2a*-D3 target sites, respectively. Similarly, we found that 70.8% (17 of 24) and 35.7% (5 of 14) of the T0 lines had heterozygous targeted mutations at the *GmFT5a*-D1 and *GmFT5a*-D2 target sites, respectively.

We subsequently used agarose gel electrophoresis and sequencing to analyze the large fragment deletions among the T0 lines transformed by combinatory dual-sgRNA/Cas9 constructs targeting *GmFT2a*-D1/D2, *GmFT2a*-D1/D3, and *GmFT5a*-D1/D2. The frequencies of the large fragment deletions were calculated by dividing the number of T0 plants with large deletions between pairs of dual-sgRNAs by the total number of T0 plants transformed by the corresponding combinatory dual-sgRNA/Cas9 vectors. We hypothesized that smaller amplicons would be generated by the deletion of a fragment between the pairs of targeted sites. The smaller DNA bands were then purified, sub-cloned, and sequenced. As expected, 15.6% (5 of 32) of the T0 lines possessed deletions induced by *GmFT2a*-D1/D2 (*GmFT2a*-D1/D2-#10, #13, #14, #16, and #32). The lengths of these deletions ranged from 599 to 1618 bp (Figure 2A). We also achieved deletion frequencies of 12.1% (4 of 33, *GmFT2a*-D1/D3-#10, #18, #25, and #32) for target fragments exceeding 4.5 kb induced by *GmFT2a*-D1/D3 (Figure 2B) and 15.8% (3 of 19, *GmFT5a*-D1/D2-#7, #13, and #14) for target fragments ranging from 1069 to 1161 bp induced by *GmFT5a*-D1/D2 (Figure 2C) among corresponding T0 events.

The frequencies of site-specific mutations at each target site were also examined, and the results are summarized in Table 2. A total of 50.0% (16 of 32) and 46.9% (15 of 32) of T0 *GmFT2a*-D1/D2 lines had targeted mutations at the *GmFT2a*-D1 and *GmFT2a*-D2 sites, respectively. Among them, 31.2% (10 of 32) simultaneously targeted both sites. Of the T0 *GmFT2a*-D1/D3 lines, 51.5% (17 of 33) and 57.6% (19 of 33) had targeted mutations at the *GmFT2a*-D1 and *GmFT2a*-D3 sites, respectively. Among them, 42.4% (14 of 33) simultaneously targeted both sites. Similarly, 63.2% (12 of 19) and 36.8% (7 of 19) of T0 *GmFT5a*-D1/D2 lines had targeted mutations at the *GmFT5a*-D1 and *GmFT5a*-D2 sites, respectively. Among them, 36.8% (7 of 19) simultaneously targeted both sites.

### 2.3. Inheritance of Large Fragment Deletion Mutations and Phenotypes of the Mutants

In this study, the seeds collected from the self-pollinated T0 lines (*GmFT2a*-D1/D2-#10, #13, #14, #16, #32; *GmFT2a*-D1/D3-#10, #18, #25, #32; and *GmFT5a*-D1/D2-#7, #13, #14) were grown in a growth chamber. To detect whether the fragment deletion mutations induced by CRISPR/Cas9 in the T0 plants could be transmitted to the progeny, we examined the genotypes of the T1 plants. We found that the fragment deletion mutations induced in T0-*GmFT2a*-D1/D2-#32 were transmitted to the T1 generation. One plant (plant 6) had heterozygous targeted mutations, and 14 of 30 plants were homozygous for large fragment deletion mutations, as shown by agarose gel electrophoresis and sequencing analysis, and all homozygous fragment deletion mutations were 1618 bp (Figure 3A). To seek ‘transgene-free’ plants, we subsequently detected two distinct regions (part of the Cas9 coding sequence and the selectable marker gene *bar*) in the T-DNA of vectors by PCR strategy and test strip, respectively. Among them, we obtained three T1 ‘transgene-free’ homozygous *ft2a* mutants (plants 10, 17, 20) with a 1618 bp deletion (Figure 3B). These results indicated that the large fragment deletion mutations were directly transmitted from the T0 plants, because the targeted deletions could not be generated de novo in the T1 plants if the transgene was absent.

The T2 progeny of these three individuals (plants 10, 17, 20), wild-type plants (WT) and homozygous *ft2a* mutants (1 bp insertion at the target site) were grown under both long-day (LD, 16 h light/8 h dark) and short-day (SD, 12 h light/12 h dark) photoperiodic conditions. We subsequently performed sequencing analysis and found that the large fragment deletion mutations were stably inherited and maintained consistent mutation types from the T1 generation to T2 generation. In addition, both T2 homozygous *ft2a* mutants (1618 bp deletion) and *ft2a* mutants (1 bp insertion) exhibited the late-flowering phenotype under SD (25.1 ± 1.2 days after emergence (DAE), *n* = 22 for WT; 31.1 ± 0.8 DAE, *n* = 18 for *ft2a* mutants with 1618 bp deletion; 30.6 ± 1.2 DAE, *n* = 20 for *ft2a* mutants with 1 bp insertion) and LD (50.7 ± 1.9 DAE, *n* = 22 for WT; 55.9 ± 1.7 DAE, *n* = 18 for *ft2a* mutants with 1618 bp deletion; 54.4 ± 1.6 DAE, *n* = 21 for *ft2a* mutants with 1 bp insertion) conditions (Figure 4). Unfortunately, the large fragment deletion mutations induced in T0-*GmFT2a*-D1/D3 and T0-*GmFT5a*-D1/D2 lines were not detected in the T1 progeny.

## 3. Discussion

The CRISPR/Cas9 system achieves genome editing by inducing DSBs at specific genomic loci, which are often repaired subsequently by NHEJ and/or HDR pathways [4]. The repair errors produced by NHEJ often generate small insertions and deletions (indels) in protein-coding regions and then disrupt target genes’ functions as a result of reading frame shifts, causing premature translational stops [5,6]. In addition, the CRISPR/Cas9 system is particularly suitable for the simultaneous mutation of multiple genes using either single sgRNA targeting identical sequence regions of homologous genes or multiple sgRNA expression cassettes targeting different loci in the genome sharing one Cas9 protein expression cassette [23,33,34,35]. However, some base indels usually lack the ability to disrupt the function of gene clusters, regulatory sequences, or functional elements in the non-coding genome, whereas the large fragment deletion method can be an effective alternative to achieve these aims. Using two sgRNAs targeting two sites within the same gene or on the same chromosome simultaneously can result in fragment deletion between the two targeted sites through the NHEJ pathway, thus providing an easier large fragment deletion method [7]. CRISPR/Cas9-mediated large fragment deletion has been reported in some plant species, such as *Arabidopsis* [21], rice [19,20], tobacco [36], cotton [23], tomato [37], maize [38], and wheat [18]. However, this technology has thus far not been reported in soybean.

The efficiency of sgRNAs directly affects the availability of CRISPR/Cas9 in soybean. We previously demonstrated that the CRISPR/Cas9 system was highly efficient in the induction of site-specific mutations in soybean, mostly consisting of small base insertions/deletions (indels), which frequently lead to gene knockouts owing to frameshift mutations that cause premature translational stops [25,32]. In this study, we first examined the mutation frequencies of five single target sites (*GmFT2a*-D1, *GmFT2a*-D2, *GmFT2a*-D3, *GmFT5a*-D1, and *GmFT5a*-D2) in T0 transgenic plants and detected frequencies of 48.0%, 52.4%, 72.2%, 70.8%, and 35.7%, respectively. We then sought to detect the ability of CRISPR/Cas9 system to generate large fragment deletions using pairs of the above five sgRNAs. Previous studies have reported that Cas9 cleaves both DNA strands between the 17th and 18th bases (3 bp from its protospacer adjacent motif (PAM) proximal end) within the target sequence [3,4,39], so we postulated that large fragment deletions would be generated between the two cutting loci and expected to obtain precise fragment deletions of 1173, 4556, and 1070 bp using the constructs *GmFT2a*-D1/D2, *GmFT2a*-D1/D3, and *GmFT5a*-D1/D2, respectively. Actually, 15.6% (5 of 32) of T0 plants produced by *GmFT2a*-D1/D2 were identified with fragment deletions ranging from 599 to 1618 bp (Figure 2A). We also achieved deletion frequencies of 12.1% (4 of 33) for target fragments exceeding 4.5 kb induced by *GmFT2a*-D1/D3 (Figure 2B) and 15.8% (3 of 19) for target fragments ranging from 1069 to 1161 bp induced by *GmFT5a*-D1/D2 (Figure 2C) among corresponding T0 events. The variation in the lengths of these deleted fragments was due to the error-prone NHEJ pathway, but they also permitted the screening of a variety of deletion mutants, a technique which holds tremendous potential for use in biotechnology applications and foundational research on protein-coding/non-coding genes or regulatory elements [2,12,20]. Our results demonstrate that CRISPR/Cas9 can be efficiently applied to produce targeted large DNA fragment deletions in soybean. Generally, a widely applied new technology has higher requirements for efficiency. As the process of large fragment deletion depends on a pair of sgRNAs to search and edit the corresponding specific sites, and the two independent DSBs should be simultaneously generated at the two loci which are subsequently rejoined through NHEJ pathway, the high efficiency of each sgRNA is essential for large fragment deletion [38].

We previously demonstrated that CRISPR/Cas9-induced mutations at specific sites in soybean could be inherited by the T1 from the T0 generation, although not all T0 lines could transmit the targeted mutations to the T1 generation, and the regularity of heritability is difficult to predict. We also found that the mutations were stably inherited and remained consistent from the T1 to the T2 generation [32]. In this study, we found that fragment deletion mutations induced in T0-*GmFT2a*-D1/D2-#32 lines were transmitted to the T1 generation. However, the large fragment deletion mutations induced in T0-*GmFT2a*-D1/D3 and T0-*GmFT5a*-D1/D2 lines were not detected in their T1 progeny plants. It is possible that in the T0 generation, CRISPR/Cas9-induced fragment deletion mutations exist only in a proportion of cells, which subsequently generate somatic sectors via cell division. If somatic mutant sectors occur in the floral primordia, the targeted mutations could be transmitted to the T1 generation through the gametes. These results indicate that the CRISPR/Cas9-induced large fragment deletions could be transmitted from the T0 to T1 generation in soybean, but its regularity is also difficult to predict. However, the large fragment deletion mutations induced by CRISPR/Cas9 were stably inherited and maintained consistent mutation types from the T1 to T2 generation.

Nowadays, the CRISPR/Cas9 system has become one of the most practical new crop breeding techniques, especially when studying important agronomical traits. Compared with CRISPR/Cas9-mediated mutations at single specific sites, large fragment deletion could eliminate not only the complete function or part of the function of protein-coding genes but also non-coding genes or long non-coding RNAs (lncRNAs) [38]. In the present study, we developed an efficient system for deleting large fragments using CRISPR/Cas9 in soybean. The *ft2a* mutants homozygous for large fragment deletions exhibited the late-flowering phenotype, indicating that this method could be utilized to study protein-coding genes. In addition, CRISPR/Cas9-mediated large fragment deletion has great potential to benefit future research on functions of non-coding genes or regulatory elements and improve agriculture via chromosome engineering or customized genetic breeding in soybean.

## 4. Materials and Methods

### 4.1. SgRNA Design and Construction of the Combinatory Dual-sgRNA/Cas9 Vectors

The sequences and other information on the endogenous soybean genes *GmFT2a* and *GmFT5a* were obtained from the Phytozome website (www.phytozome.net). All sgRNAs for each single target site of *GmFT2a* and *GmFT5a* were designed using the web-based tool CRISPR-P (http://crispr.hzau.edu.cn/CRISPR2/) [40]. We selected five sgRNAs targeting *GmFT2a* and *GmFT5a* sites named *GmFT2a*-D1, *GmFT2a*-D2, *GmFT2a*-D3, *GmFT5a*-D1, and *GmFT5a*-D2. The corresponding CRISPR/Cas9-based vectors were constructed based on our previous study [32]. To delete the targeted fragments and improve transformation efficiency, pairs of sgRNA cassettes and one Cas9 cassette were assembled together in each vector. To generate the *GmFT2a*-D1/D2 and *GmFT2a*-D1/D3 constructs, the construct *GmFT2a*-D1 was digested by the restriction enzymes *Asc*I and *Spe*I. The products were analyzed using 1% agarose gel electrophoresis. The smaller DNA band (approximately 590 bp) was extracted and inserted into the purified *GmFT2a*-D2 and *GmFT2a*-D3 constructs digested by the restriction enzymes *Asc*I and *Avr*II. The *GmFT5a*-D1/D2 construct was assembled according to the same principle. Subsequently, the five vectors targeting single sites and the three combinatory dual-sgRNA/Cas9 vectors were individually electroporated into *Agrobacterium tumefaciens* strain EHA105. The *Agrobacterium tumefaciens*-mediated transformation of the soybean cultivar Jack was performed according to a previously described protocol [41].

### 4.2. Determination of the Site-Specific Mutation Frequencies at Each Target Site and Large Fragment Deletions

Genomic DNA from the leaves of transgenic soybean plants was extracted using the Plant Genomic DNA Kit (Cwbiotech, Beijing, China) and subsequently used for PCR with specific primers (Appendix A). The frequencies of the site-specific mutations at each target site were determined according to our previous study [32]. To detect the frequencies of large fragment deletions, the forward primers were designed using the upstream region of the front sgRNA targets, and the reverse primers were designed in the downstream region of the postpositional sgRNA targets. The PCR products were analyzed via 1% agarose gel electrophoresis. The lengths of the PCR products amplified by primer 11/primer 12 and primer 15/primer 16 were predicted to be 1959 and 1556 bp, respectively. Some shorter bands were generated due to targeted fragment deletions. The DNA of these shorter bands was recovered using a Zymoclean Gel DNA Recovery Kit (Zymo research, Los Angeles, CA, USA) and cloned using a pClone007 Blunt Simple Vector Kit (Tsingke, Beijing, China). Ten colonies per DNA clone were then sequenced for genotyping and analyzed by sequence alignment to the wild-type sequence. The length of the PCR product amplified by primer 13/primer 14 was 5448 bp. Because some non-specific bands may be produced during the PCR amplification of long fragments, an alternative method was adopted: the extension time of the PCR program was greatly shortened and was only sufficient to amplify segments of less than 1 kb. In this study, the sequences between primer 13 and primer 14 were short enough to be amplified if they contained targeted large fragment deletions. The DNA recovered from the target bands was sub-cloned, and individual colonies were sequenced to reveal the clonal genotypes, as mentioned above.

### 4.3. Flowering Time Measurements and Statistical Analyses

To seek ‘transgene-free’ plants among mutants with large fragment deletions, we detected two distinct regions (part of the Cas9 coding sequence and the selectable marker gene *bar*) in the T-DNA of vectors by PCR strategy (the primer sequences are listed in Appendix A) and test strip, respectively. To gather and compare the phenotype of flowering time, the T2 ‘transgene-free’ homozygous *ft2a* mutants (1618 bp deletion), WT plants, and the homozygous *ft2a* mutants (1 bp insertion at target site *GmFT2a*-SP2) that we previously reported [32] were grown under LD (16 h light/28 °C and 8 h dark/20 °C) and SD (12 h light/28 °C and 12 h dark/20 °C) photoperiodic conditions. The flowering time of each plant was recorded as days from emergence to the R1 stage, the first flower’s appearance at any node in the main stem [42]. Statistical analyses were performed using Microsoft Excel, and the significance of the differences among control and treatments was analyzed by ANOVA test, followed by Tukey–Kramer at a significant level of 1%. SigmaPlot 10.0 was used for drawing boxplots and bar plots. The flowering time is shown as the mean values ± standard deviation.

## 5. Conclusions

In this study, we developed an efficient system for deleting large fragments in soybean using CRISPR/Cas9, and succeeded in targeting *GmFT2a* and *GmFT5a*; this system could benefit future research on gene function and improve agriculture via chromosome engineering or customized genetic breeding in soybean.

## Figures and Tables

**Figure 1 ijms-19-03835-f001:**
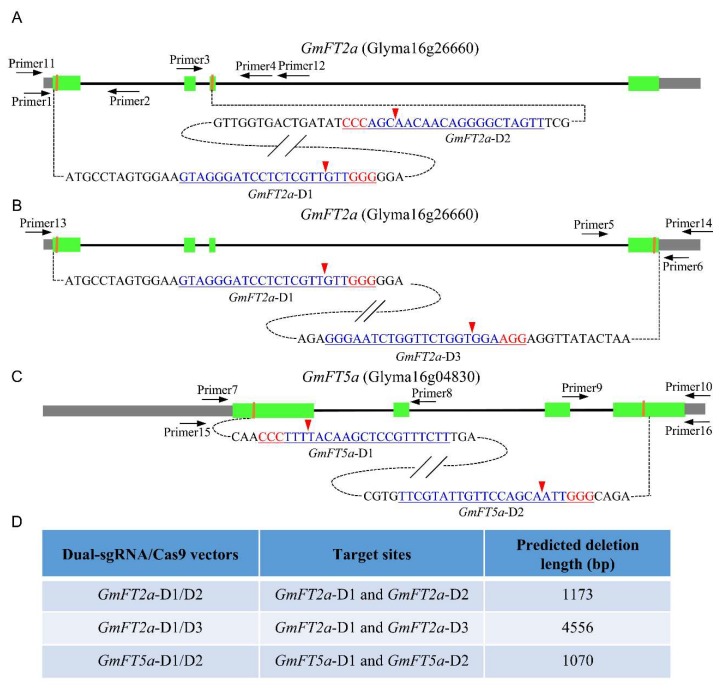
Gene structures of *GmFT2a* and *GmFT5a* with target sites of CRISPR/Cas9. Green stripe, exon; black solid line, intron; gray stripe, UTR (untranslated region). (**A**,**B**) *GmFT2a*-D1, *GmFT2a*-D2, and *GmFT2a*-D3 were target sites of CRISPR/Cas9 designed in the first, third, and fourth exon of *GmFT2a*, respectively. (**C**) *GmFT5a*-D1 and *GmFT5a*-D2 were target sites of CRISPR/Cas9 designed in the first and fourth exon of *GmFT5a*, respectively. Red letter, PAM (protospacer adjacent motif). The red arrowheads indicate the cleavage sites of the CRISPR/Cas9 system. Primer1–16 are specific primers used for PCR. (**D**) Combinatory dual-sgRNA/Cas9 vectors for targeted fragment deletions of *GmFT2a* and *GmFT5a*.

**Figure 2 ijms-19-03835-f002:**
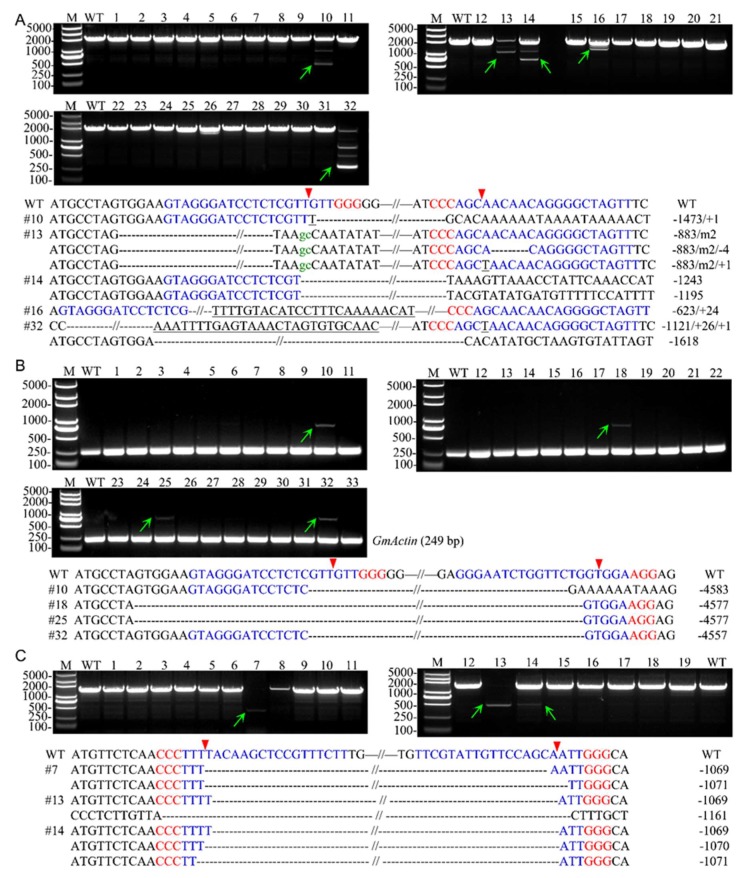
Large fragment deletions induced by the CRISPR/Cas9 system in the T0 line soybean plants. (**A**,**B**,**C**) Agarose gel electrophoresis and sub-clone sequencing results of the PCR products amplified from the genomic DNA of T0 soybean plants transfected with the constructs *GmFT2a*-D1/D2, *GmFT2a*-D1/D3, and *GmFT5a*-D1/D2, respectively. Nucleotides in blue represent the 20 bp gRNA spacer sequences. Nucleotides in red represent PAM (protospacer adjacent motif). Red triangles are the putative cleavage sites; green arrowheads indicate the lines which were subsequently sequenced. Dash, base deletion; underline, base insertion. Green lowercase letter, base mismatch. The lengths of the insertions and/or deletions are presented in the right column. M, DL2000 Plus ladder DNA marker; WT, wild type. *GmActin* was used as an internal control gene in (**B**) to show that the DNA was of good quality for PCR.

**Figure 3 ijms-19-03835-f003:**
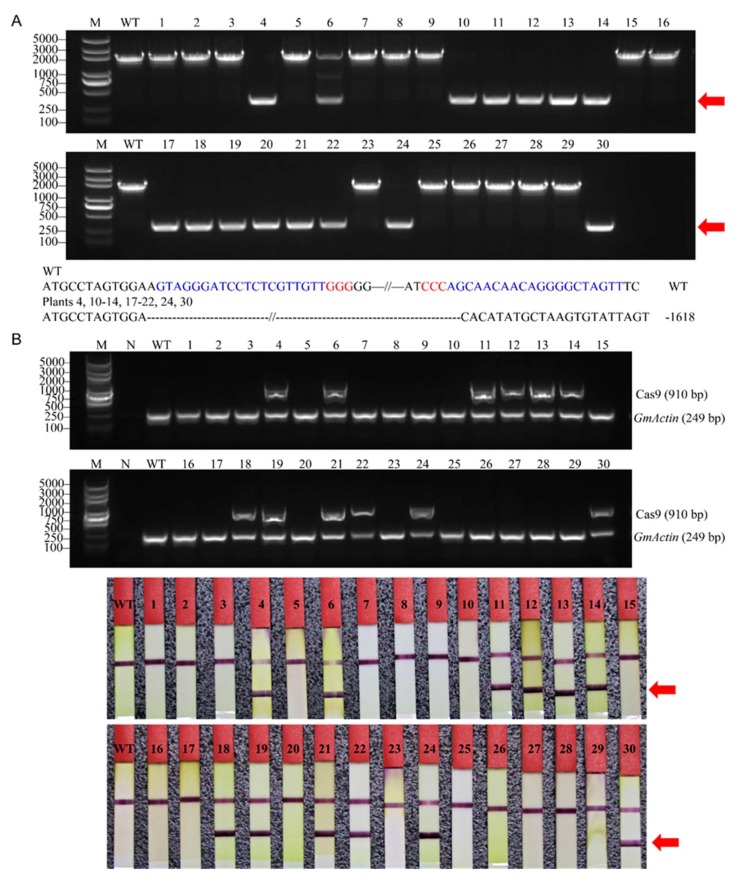
Large fragment deletion mutations induced in T0-*GmFT2a*-D1/D2-#32 lines were transmitted to the T1 generation. (**A**) M, DL2000 ladder DNA marker; WT, wild type; dash, base deletion. Red arrowheads indicate the lines which were subsequently sequenced. Plants 4, 10–14, 17–22, 24, and 30 were homozygous for large fragment deletion mutations, and all homozygous fragment deletion mutations were 1618 bp. (**B**) Identifying ‘transgene-free’ plants. Upper part, gel image of PCR products. Cas9 (910 bp), part of the Cas9 coding sequence. *GmActin* was used as an internal control gene. M, DL2000 Plus ladder DNA marker. N, negative control (water as template). Bottom half, detection of the selectable marker gene *bar* by test strip. The bands at the red arrowhead indicate that *bar* is positive. WT, wild-type plant. Labels 1–30, individual plants of the T1 progeny of the T0-*GmFT2a*-D1/D2-#32 line.

**Figure 4 ijms-19-03835-f004:**
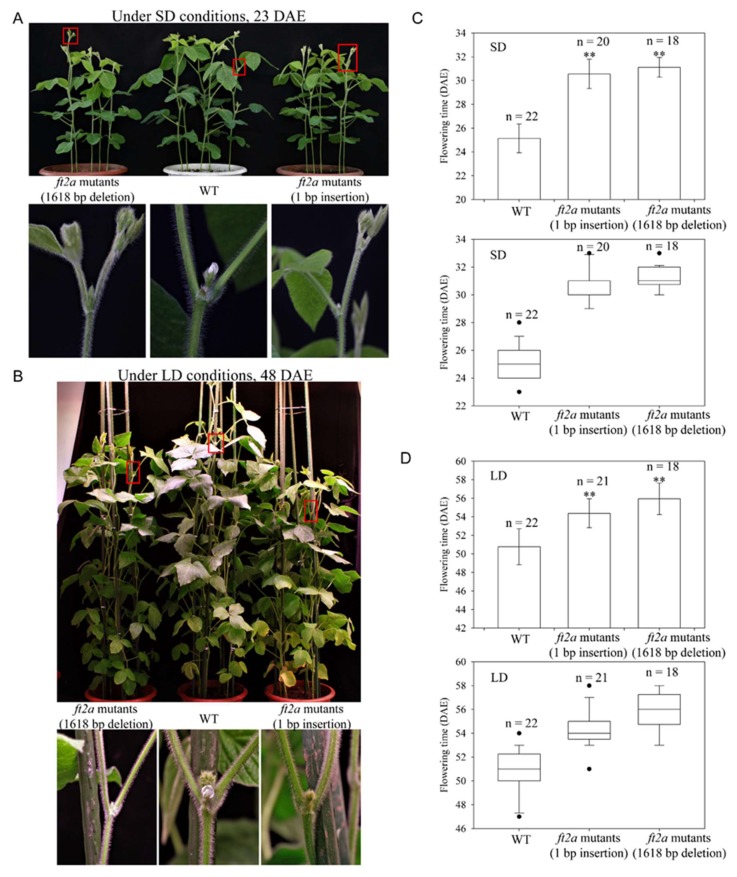
The *ft2a* mutants homozygous for large fragment deletions exhibited late flowering under both long-day (LD) and short-day (SD) conditions. (**A**,**B**) Flowering phenotypes of the *ft2a* mutants (1618 bp deletion), WT plants, and the *ft2a* mutants (1 bp insertion) under SD and LD conditions, respectively. Red box, magnified view. DAE, days after emergence. (**C**,**D**) Boxplots and bar plots indicating flowering time of the *ft2a* mutants (1618 bp deletion), WT plants, and the *ft2a* mutants (1 bp insertion) under SD and LD conditions, respectively. *n*, exact number of individual plants identified. The flowering time is shown as the mean values ± standard deviation. ** both *ft2a* mutants (1618 bp deletion) and the *ft2a* mutants (1 bp insertion) exhibit highly significant late flowering (*p* < 0.01).

**Table 1 ijms-19-03835-t001:** Frequencies of mutations at target sites among T0 transgenic plants transformed by constructs targeting single sites.

Target Sites	No. of Transgenic Events Identified	No. of Events with Targeted Mutations	Mutation Frequency
*GmFT2a*-D1	25	12	48.0%
*GmFT2a*-D2	21	11	52.4%
*GmFT2a*-D3	18	13	72.2%
*GmFT5a*-D1	24	17	70.8%
*GmFT5a*-D2	14	5	35.7%

**Table 2 ijms-19-03835-t002:** Frequencies of the site-specific mutations at each target site and large fragment deletions among the T0 lines transformed by constructs targeting *GmFT2a*-D1/D2, *GmFT2a*-D1/D3, and *GmFT5a*-D1/D2.

Constructs	No. of Transgenic Plants Identified ^a^	No. of Plants with no Mutation	No. of Plants with Targeted Mutations at the First Site	No. of Plants with Targeted Mutations at the Second Site	No. of Plants with Simultaneous Mutations at the Two Target Sites	No. of Plants with Large Fragment Deletions
*GmFT2a*-D1/D2	32	11 (34.4%)	16 (50.0%)	15 (46.9%)	10 (31.2%)	5 (15.6%)
*GmFT2a*-D1/D3	33	11 (33.3%)	17 (51.5%)	19 (57.6%)	14 (42.4%)	4 (12.1%)
*GmFT5a*-D1/D2	19	7 (36.8%)	12 (63.2%)	7 (36.8%)	7 (36.8%)	3 (15.8%)

^a^ No. of transgenic plants identified = No. of plants with targeted mutations at the first site + No. of plants with targeted mutations at the second site − No. of plants with simultaneous mutations at the two target sites + No. of plants with no mutation.

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
