# Peer review of "CRISPR/Cas9-Mediated Deletion of Large Genomic Fragments in Soybean"

_ijms, 2018, doi:10.3390/ijms19123835_

Reviewer 1 Report

The authors describe a successful knock out of two homologs of the FLOWERING LOCUS in soybean using CRISPR/Cas9 transduction. They used a dual sgRNA strategy to introduce large deletions in the respective genes. Using seeds from the genome edited plants, they identified one plant were the knock out was inherited and the transgenically introduced Cas9 had not been transmitted.

The experiments are planned and described well, the results are plausible and encouraging. THe manuscript is very well written. The authors may only add a comment on the overall efficacy of FLOWERING LOCUS knock-out soybeans using their method. I advise to accept the manuscript with only minor modifications.

Figure 2: Figures with increased contrast should be added for the lanes with weak mutation bands. Some of these bands are barely visible.

Lines 164ff:

The enumeration is confusing. Please indicate how many (or %) of the lines have mutations on one and how many have mutations on both target sites. The same holds true for table 2. In the end one should come up with 100%. Otherwise it is hard to conceive the percentage / absolute number of plants.

Lines 230f:

Please rephrase “are usually lack of abilities” to “usually lack the ability”

Author Response

Thank you very much for your thoughtful consideration of our manuscript, “CRISPR/Cas9-mediated deletion of large genomic fragments in soybean” (ijms-386100), and your insightful comments. We have carefully revised the manuscript according to these comments. Our responses are listed below.

Figure 2: Figures with increased contrast should be added for the lanes with weak mutation bands. Some of these bands are barely visible.

Response: We have replaced the Figure 2 with a better one.

Lines 164ff:

The enumeration is confusing. Please indicate how many (or %) of the lines have mutations on one and how many have mutations on both target sites. The same holds true for table 2. In the end one should come up with 100%. Otherwise it is hard to conceive the percentage / absolute number of plants.

Response: We added the percentage number of plants in Line 165-172.

Formula

No. of transgenic plants identified = No. of plants with targeted mutations at the first site + No. of plants with targeted mutations at the second site - No. of plants with simultaneous mutations at the two target sites + No. of plants with no mutation

Lines 230f:

Please rephrase “are usually lack of abilities” to “usually lack the ability”

Response: We have replaced the“are usually lack of abilities” to “usually lack the ability”. Line 233-234.

Reviewer 2 Report

Article title

CRISPR/Cas9-mediated deletion of large genomic fragment I soybean.

Cai, Y., Chen, L., Sun, S., Wu, C., Yao, W., Jiang, B., Han, T., Hou,W.

General comments;

This study aimed to verify whether large deletion mutations can be induced by using dual-sgRNA/Cas9 system in two FT loci of Glycine max. Authors showed very clear evidences that large fragment deletions could be generated with high frequencies (>12%) in both loci (Table 2). They simultaneously showed in the current trial that the transmission of the deletion mutation to the next generations was not easy event, only one T0 plant (#32) was successful. This reviewer thinks that the results shown in the manuscript are very clear and interesting, and agree this paper is acceptable for publication.

Specific comments;

However, this reviewer would like to ask the authors to deepen more the discussion about the causes how transmission rate became lower than expected (since, mutation rate in T0 generation was very high). Are there some explainable causes? For example, a possibility that off-target event affected the transmission rate.

They did not mention any about off-target effect in this paper. Did authors make sure the current constructs do not have any off-target effects, or already checked in previous works? If available, authors should mention this point in the current manuscript.

Description of “ by a one-way analysis of variance least significant difference test (LSD) at the 0.01 probability level” Line 333-334 (11 of 14) is wrong. This can be “ the significance of difference among control and treatments was analyzed by ANOVA test, followed by LSD at significant level of 1%.” But this reviewer does not recommend using LSD test for the multiple comparisons of the means. LSD (Fishers) test is similar to a set of individual t-test, and is incorrect. Avoiding LSD test is recommendable, use other statistical test instead, eg. Tukey-Kramer etc.

Author Response

Thank you very much for your thoughtful consideration of our manuscript, “CRISPR/Cas9-mediated deletion of large genomic fragments in soybean” (ijms-386100), and your insightful comments. We have carefully revised the manuscript according to these comments. Our responses are listed below.

Specific comments;

However, this reviewer would like to ask the authors to deepen more the discussion about the causes how transmission rate became lower than expected (since, mutation rate in T0 generation was very high). Are there some explainable causes? For example, a possibility that off-target event affected the transmission rate.

Response: In this study, we used the Agrobacterium tumefaciens-mediated transformation. Most T0 soybean transgenic plants are usually chimera. It is possible that in the T0 generation, CRISPR/Cas9-induced fragment deletion mutations exist only in a proportion of cells, which subsequently generate somatic sectors via cell division. If somatic mutant sectors occur in the floral primordia, the targeted mutations could be transmitted to the T1 generation through the gametes. So the transmission rate from the T0 to T1 generation is difficult to predict. In contrast, the large fragment deletion mutations induced by CRISPR/Cas9 were stably inherited and maintained consistent mutation types from the T1 to T2 generation. Line 276-283.

They did not mention any about off-target effect in this paper. Did authors make sure the current constructs do not have any off-target effects, or already checked in previous works? If available, authors should mention this point in the current manuscript.

Response: Off-target events are notable in the application of CRISPR/Cas9. However, it may not be a fatal problem in plant basic research. The risk of off-target events induced by CRISPR/Cas9 in plants may not be high on account of the lower somatic mutation frequency in tissue culture-based transformation or other mutagenic treatments. Through careful target selection, potential off-target mutations could be minimized. In addition, undesired off-target mutations in plants could be eliminated by hybridization. In short, the adverse effects of off-target events could be eliminated in plant basic research.

Description of “ by a one-way analysis of variance least significant difference test (LSD) at the 0.01 probability level” Line 333-334 (11 of 14) is wrong. This can be “the significance of difference among control and treatments was analyzed by ANOVA test, followed by LSD at significant level of 1%.” But this reviewer does not recommend using LSD test for the multiple comparisons of the means. LSD (Fishers) test is similar to a set of individual t-test, and is incorrect. Avoiding LSD test is recommendable, use other statistical test instead, eg. Tukey-Kramer etc.

Response: We have revised this part according to your kind comments. Line 341-343.